# The Impact of Stress, Microbial Dysbiosis, and Inflammation on Necrotizing Enterocolitis

**DOI:** 10.3390/microorganisms11092206

**Published:** 2023-08-31

**Authors:** Venkata Yeramilli, Riadh Cheddadi, Heather Benjamin, Colin Martin

**Affiliations:** Division of Pediatric, Department of Surgery, University of Alabama at Birmingham, 1600 7th Ave. S., Lowder Building Suite 300, Birmingham, AL 35233, USA

**Keywords:** necrotizing enterocolitis, prenatal stress, cortisol, inflammation, microbiome

## Abstract

Necrotizing enterocolitis (NEC) is the leading cause of intestinal morbidity and mortality in neonates. A large body of work exists; however, the pathogenesis of NEC remains poorly understood. Numerous predictors have been implicated in the development of NEC, with relatively less emphasis on maternal factors. Utilizing human tissue plays a crucial role in enhancing our comprehension of the underlying mechanisms accountable for this devastating disease. In this review, we will discuss how maternal stress affects the pathogenesis of NEC and how changes in the intestinal microbiome can influence the development of NEC. We will also discuss the results of transcriptomics-based studies and analyze the gene expression changes in NEC tissues and other molecular targets associated with the pathogenesis of NEC.

## 1. Introduction

Necrotizing enterocolitis (NEC) is the leading cause of intestinal morbidity and mortality among premature infants and is characterized by the acute onset of inflammation and necrosis of the intestine [1,2]. Because the pathogenesis of NEC is not fully clear and is thought to be multifactorial, it is difficult to predict which infant will develop NEC. In contrast to the many suspected risk factors, there are only a few clearly associated risk factors: prematurity, low birth weight, enteral feeding, and neonatal infection and inflammation [3,4,5]. Most interventions for NEC in newborns are targeted after birth because of the prevailing view that NEC develops in response to dietary and microbial factors during the post-natal period [6]. Emerging evidence challenges this view and raises the possibility that maternal and environmental factors in utero regulate maternal–fetal signals and can modulate the development and pathogenesis of NEC [7,8]. In this review, we discuss some of these factors based on data from our laboratory and others while underscoring biomarkers and molecular targets that can be employed to decrease the incidence and severity of the disease.

## 2. Maternal Stress and Birth Outcomes

Clinically, psychological stress is one of the underappreciated causes of reproductive fragility in women. Stress during gestation decreases the offspring’s birth weight and increases the likelihood of prematurity, a known predictor of NEC [9,10]. Further, maternal stress or anxiety during pregnancy can have long-term effects on the development of neurological and inflammatory diseases in the offspring [11,12].

Many psychological stressors (Table 1) impact a system’s biology during pregnancy. The response to these stressors is influenced by the nature, timing, and duration of the stressors, as well as a person’s inherent susceptibility to stress and personal characteristics, all of which influence the way in which an individual copes with stress. The endocrine hypothalamic-pituitary-adrenal (HPA) axis and the locus ceruleus-norepinephrine (LC/NE) nervous system are key mediators. In response to a stressor, norepinephrine (NE), epinephrine (EPI), and catecholamines are released, followed by the activation of the HPA axis [13]. Components of the central nervous system include LC/NE, parvicellular corticotropin-releasing hormone (CRH), arginine-vasopressin (AVP), CRH neurons, and other catecholaminergic cell groups. AVP and CRH activate the pituitary adrenocorticotrophic hormone (ACTH), resulting in the production of glucocorticoids [14]. By means of a negative feedback loop, glucocorticoids exert control on the hypothalamic-pituitary axis and secretion of ACTH, CRH, and AVP [15]. 

During gestational stress, placental CRH gradually increases in maternal circulation during the third trimester, resulting in elevated cortisol levels that suppress maternal CRH secretions [16]. By activating the embryonic HPA axis, placental CRH may initiate labor. Elevated placental CRH concentrations have been linked to preterm labor [17,18]. Furthermore, preterm fetuses have higher plasma EPI concentrations than term fetuses [19]. This is most likely mediated by elevated glucocorticoids which convert NE to EPI via the enzymatic activation of phenylethanoalamine-N-methyltransferase (PNMT) [15]. Additionally, maternal stress during pregnancy is also associated with the secretion of catecholamines by the sympathetic nervous system [20]. Depression and anxiety during the first trimester of pregnancy are risk factors for the development of preeclampsia in the second and third trimesters [21,22,23]. From 18 to 20 weeks of gestation, preeclamptic mothers have elevated serum concentrations of placental CRH [24]. Evidence suggests that stress-induced preeclampsia may be an important risk factor because the incidence and severity of NEC in premature infants born to preeclamptic mothers are significantly higher [25]. In addition, there is evidence that excessive glucocorticoids due to sustained endogenous embryonic cortisol production or maternal administration of synthetic corticosteroids may lead to intrauterine growth restriction (IUGR) [26]. IUGR is more prevalent in preterm infants and is associated with greater rates of NEC [25,27,28]. In conclusion, maternal stress disrupts the embryonic glucocorticoid environment and is linked to negative birth outcomes. Potential markers for identifying women at risk for preterm birth should include the assessment of maternal cortisol and CRH concentrations and the quantification of psychological stress in expectant women using validated questionnaires. 

## 3. Paternal Stress and Birth Outcomes

The impact of paternal stress on neonatal gut health has received relatively less attention compared to maternal stress in the existing literature. Recent studies have highlighted how parental stress can affect the offspring’s quality of life through stress-induced epigenetic changes in sperm [29]. These changes can be influenced by environmental factors or stressors and can have long-term effects on gene expression and cellular function, contributing to the development of complex chronic diseases [30]. We believe that the significance of paternal epigenetic alterations is underestimated, and the lifestyle of the father, particularly in high-stress environments, plays a crucial role in programming an infant’s epigenome. This programming may contribute to the development of complex neonatal diseases like NEC. Additionally, we hypothesize that this concept is also linked to an increased susceptibility to neonatal microbiome dysbiosis and gut injury. Understanding how parental stress exposure is communicated to the developing offspring’s gut is crucial for unraveling the causes of intestinal dysbiosis or injury.

Previously, it was believed that spermatozoa DNA was static and unaffected by external influences [29]. However, recent research challenges this paradigm and shows that mature spermatozoa are susceptible to various challenges during their maturation in the epididymis, including stress, trauma, dietary disruptions, and drug abuse. Epidemiological investigations have provided compelling evidence linking stress exposures in males with disease risk in subsequent generations [31,32]. Studies on the Swedish Famine of 1836 and a cohort of Holocaust survivors and their offspring have demonstrated associations between early childhood food supply and adverse health outcomes in sons and grandsons, as well as alterations in glucocorticoid receptors in the offspring of survivors [33,34,35]. Studies on rodents have also shown that germ cells are susceptible to stressful environments across the paternal lifespan. Manipulations such as increased stress exposure or administration of stress hormones prior to mating have resulted in modified behavior and various cellular and molecular changes in the offspring [36,37,38]. A recent study using the intruder–resident method, electric shock, and food deprivation in male rats before mating revealed increased expression of oxytocin and vasopressin genes in the offspring [39]. These results indicate that paternal stress modulates the hypothalamic-pituitary-adrenal (HPA) axis and leads to increased glucocorticoid signaling in the offspring, factors which are known to contribute to microbiome dysbiosis [40].

Collectively, these findings from human and animal studies suggest a possible relationship between paternal stress and neonatal intestinal injury. While human studies exploring the molecular profiles of stress in the male germline are still incomplete, conducting such research could provide valuable insights into previously unexplored pathways of intergenerational inheritance resulting from paternal stress exposures.

## 4. Microbial Dysbiosis in NEC

Even though the pathophysiology of NEC remains elusive, multiple lines of evidence indicate that the disruption of host–microbiota interactions plays a crucial role in the development of NEC [1]. In utero, when the intestine has not yet been colonized by commensals, there is no incidence of neonatal enterocolitis [41]. In premature infants, antibiotic use and substitution of breast milk with formula during the first week of life disrupt the colonization of commensal bacteria and are strongly associated with the development of NEC [1]. Evidence from animal studies suggests that alterations in gut microbial communities and intestinal immaturity are crucial factors in mediating tissue injury in the gastrointestinal tract [42,43]. *Bifidobacterium*, *Bacteroidota*, *Escherichia*, and *Parabacteroides* dominate the gastrointestinal communities of newborn healthy babies [44,45]. Thereafter, the diversity of the gut microbiota increases swiftly and is shaped by factors such as diet and antibiotic exposure [44,45]. In contrast, Bacilli initially dominates the bacterial community in premature infants, followed by *Gammaproteobacteria*, such as *Klebsiella*, *Escherichia*, and other Enterobacteriaceae [46]. At birth, the gut microbiota of preterm infants is compositionally distinct and less diverse than that of full-term infants [47,48]. After hospital discharge, however, the preterm gut microbiota swiftly diversifies and becomes indistinguishable from those of full-term infants by 2 years of age [49,50]. 

More than two decades ago, Claud et al. proposed that the lack of bacterial diversity could lead to the development of NEC [51]. Since then, several studies have documented the absence of microbial diversity in premature and NEC infants [52,53,54]. The microbial signature of NEC is characterized by an increase in *Pseudomonadota*, especially *Enterobacteriaceae*, and a decrease in *Bacillota* and *Bacteroidota* [55]. As the predominant Gram-negative bacterial group in the intestine of preterm infants, it is believed that *Pseudomonadota* stimulate proinflammatory immune responses via TLR4 signaling, and lead to the loss of barrier function, ischemia, and tissue necrosis in preterm infants [55]. Recently, Ji et al. investigated the effect of exogenous Autoinducer-2 (AI-2) on intestinal dysbiosis and inflammation in a neonatal mouse model of necrotizing enterocolitis [56]. The addition of AI-2 to the formula milk of NEC mice reduced intestinal injury scores and the expression of proinflammatory factors. The research shows that AI-2 partially reverses flora disorder in the NEC mouse model, with increased the abundance of *Pseudomonadota* and decreased the abundance of *Bacteroidota* at the phylum level [56]. At the genus level, the study reveals that *Helicobacter* and Clostridium_sensu_stricto_1 have significantly greater abundance in the NEC group, while Lactobacillus shows the opposite trend. Additionally, the abundances of *Klebsiella*, *Rodentibacter*, and *Enterococcus* are significantly higher in the AI-2–treated group compared to the NEC and control groups [56]. Thus, modifying the formula milk can be used as a potential therapeutic approach to treat NEC.

Butyrate, a short-chain fatty acid produced by the gut microbiota, has been shown to have anti-inflammatory and protective effects on the intestinal mucosa [57]. Butyrate has been implicated in the pathogenesis of NEC. Wang et al. showed that butyrate induces development-dependent necrotizing enterocolitis-like intestinal epithelial injury via necroptosis [58]. In contrast, Lie et al. reported beneficial effects of butyrate in intestinal injury [59]. The authors showed that high-dose butyrate upregulated inflammatory marker IL-6, while low-dose butyrate protected cells from injury by reducing IL-6 expression. Further, administration of butyrate in NEC mice reduced intestinal damage, reduced IL-6 and NF-ĸB expression, and improved barrier function [59]. While these results are promising, the specific role of butyrate in the development and progression of NEC is still under investigation and requires further research.

A greater taxonomic and genomic resolution is required to further enhance the predictive value of microbial signatures in NEC. To analyze microbial communities, 16s ribosomal RNA sequencing has been utilized in most investigations to date. Utilizing more advanced technologies, such as whole genome shotgun sequencing, will lead to the identification of additional principal microorganism candidates involved in the pathogenesis of NEC [60]. 

## 5. Maternal Stress-Induced Changes in Microbiota Composition

In addition to environmental factors that influence the interactions between host and microbiome in preterm neonates, maternal stress can contribute to the development of necrotizing enterocolitis by disrupting the microbial communities of both mother and child. We and other labs have demonstrated that psychological stress in pregnant rodents causes imbalances in their offspring’s intestinal microbiota [42,61,62]. In addition, prenatal stress alters the composition of the maternal gastrointestinal and vaginal microbiota, suggesting that the dysbiotic microbiota in the offspring are likely inherited from the mothers [63]. Similar associations between maternal psychological stress during pregnancy and gut dysbiosis have been observed in humans as well [64]. Furthermore, studies indicate prenatal stress also impacts the expression of genes involved in inflammation and immune cell recruitment in the fetal intestine [61]. Interestingly, dysbiotic maternal vaginal microbiota induced by stress have been found to contribute in part to changes in gene expression in the hypothalamus of male progeny, particularly in response to stress exposure as an adult [61]. In addition, adult male progeny of mothers who experienced prenatal stress produce higher levels of corticosterone in response to stressful situations compared to adult male offspring of mothers who did not experience prenatal stress [62]. These results indicate that the transmission of stress-induced dysbiotic maternal microbiota to newborns disrupts the HPA axis, thereby influencing the stress responses of the offspring throughout their lifetimes. In adults, non-pregnant mice chronic stress exposure led to a significant alteration in the microbial composition, a decrease in the abundance of lactobacilli, and an increase in *Proteobacteria* and *Escherichia coli* [65,66,67,68]. In experimental animals, anxiety-like behavior, an increase in pro-inflammatory cytokines, and recruitment of monocytes to the colon was observed following the transplantation of fecal microbiota from stressed to non-stressed mice [65]. These effects were not observed in recipients of non-stressed mice’s fecal microbiota. Mechanism-wise, it could likely be that stress hormones may directly affect microbial proliferation and thereby influence composition and diversity in the gut [69]. Additionally, the microbial communities could be impacted by stress-induced changes in gastrointestinal physiology, including alterations in the profiles and levels of secretory IgA and short-chain fatty acids, and a decrease in the production of bile and gastric acid [42,70,71,72,73]. 

## 6. Gene Expression Profiling in NEC

The major risk factors for NEC’s development are preterm birth, aberrant bacterial colonization, and enteral feeding [1]. Of these, prematurity is considered the established risk factor. One of the serious challenges neonatologists’ faces is determining accurate or early clinical signs and symptoms of NEC. Due to its abrupt onset, NEC is often detected only at the advanced stages [74]. Therefore, an early diagnosing method for recognizing at risk preterm babies is needed. Several attempts have been made to identify key genes expressed in NEC tissues or distinguish them from related pathologies. Despite these efforts, no suitable biomarkers have been identified so far. In this section, we review some of the recent “omics” approaches that have been undertaken to identify key genes and pathways associated with NEC (Table 2). We will also discuss how some of these genes modulate microbiome and stress signals that could impact NEC.

Using a microarray approach, Kathy et al. published the first comprehensive database on differential gene expression profiles in NEC tissues compared to surgical controls with a non-inflammatory intestinal condition [75]. They observed significant changes in the expression of genes involved in multiple pathways of angiogenesis, arginine metabolism, cell adhesion, chemotaxis, extracellular matrix remodeling, hypoxia, oxidative stress, and inflammation. The key dysregulated genes were TLR2, TLR4, and TREM1 which are mediated via NF-kB, AP-1, and HIF1A transcription factor pathways [75]. These findings indicate predominant microbial and inflammatory involvement in NEC. Building on these observations, Chen et al. used a similar microarray approach to identify a set of key genes that could provide insights into the molecular mechanisms of the development and progression of NEC [76]. Using q-PCR, the authors found the expression of IL-8 was upregulated in NEC compared to controls. IL-8 is known to regulate inflammatory responses through the recruitment of neutrophils to tissues [77]. Consistent with the microarray and qPCR data, Weitkamp et al. showed that IL-8 mRNA and protein were overexpressed in the intestinal tissue and serum, respectively, in NEC babies compared to healthy controls [78]. These findings suggest IL-8 can be used as a potential biomarker. Besides IL-8, ATG, KNG, ACAB, and CAT expressions in NEC, small intestine tissues were lower than controls [76]. Notably, ACAB and CAT are associated with propionate and tryptophan metabolic pathways and oxidative stress [79,80,81]. The encoding catalase in CAT converts reactive oxygen species to water and oxygen and eventually plays a role in oxidative stress pathways [82]. Thus, the reduced expression of CAT in NEC may trigger the development of NEC through the weakening of the antioxidant stress pathway [76]. Further studies on the genes identified in this study are needed to understand their role in NEC. We performed a similar pilot microarray analysis using RNA isolated from paraffin banked specimens from babies with NEC and aged matched controls from infants with jejunal and ileal atresia. We identified several genes that exhibited ≥ two-fold changes in NEC compared to controls (Table 2). Pathway analysis identified the signal transducer and activator of transcription 3 (STAT3), Prolactin (PRL), interleukin-1 beta (IL-1β), signal transducer and activator of transcription 1(STAT1), and Interferon-gamma (IFNγ) as the top canonical factors. These findings are consistent with the predicted involvement of an inflammatory cascade in regulating NEC. Patients with STAT1/STAT3 signaling defects exhibit an imbalance in the microbiome [83]. Further, the diversity of gut microbiota at weaning is altered in PRL receptor-null mice [84], suggesting some of these NEC-induced genes also modulate the microbiome, which could indirectly lead to an inflammatory environment.

The development of high-throughput sequencing of RNA transcripts (RNA-Seq) has become a method of choice for transcriptional profiling of differentially expressed genes. Eric et al. reported the first RNA-Seq–based gene expression profiling in NEC [85]. The genes identified (Table 2) showed a remarkable similarity to those implicated in Crohn’s disease. Notably, the increased expression of CXCL8, CCL10, and IL-8 in NEC agrees with previous reports [76,85,86]. Elevated levels of these cytokines were also reported in preterm babies diagnosed with NEC [87]. In line with the role of abnormal bacterial colonization in NEC, the researchers also noted the upregulation of TLR4 and TLR10 [88]. Similar findings were reported by Chan et al., who performed mRNA sequencing on intestinal tissues from surgically treated NEC cases, and by Ng et al., who investigated microRNAs (miR) in NEC [75,89]. Both studies identified TLRs as a crucial pathway in NEC’s development. In addition, a more recent RNA-Seq investigation identified significant enrichment of differentially expressed genes in the TLR signaling pathway and cytokine–cytokine receptor interactions [90]. In the past decade, several human and animal studies have shed light on the association of TLR4 with NEC [91,92]. In addition to TLRs, Rabiul et al.’s findings further supported the involvement of the immune system and complement cascade as key mechanisms in the pathogenesis of NEC [93]. Previous studies have indicated that premature infants have lower amounts of C3 and C9 proteins [94,95]. It is believed that increased levels of C5a in the serum of babies with NEC contribute to the initiation of inflammation [96]. In our study, we observed heightened levels of C3 in the small intestine using a mouse model that mimics NEC-like injury [43]. Taken together, these results highlight the importance of TLR and inflammation in NEC and provide evidence that “omics”-based approaches can identify potential genetic pathways in NEC. There is also evidence linking TLR and stress reactivity. TLRs can stimulate the release of steroids, including cortisol, from the adrenal glands [97]. Another study found that cortisol exposure influenced the gene expression of TLR-5 in rainbow trout embryos [98]. In a more recent study, early and recent life exposure to stress is associated with greater ex-vivo inflammatory responsivity to stimulation by TLR-2 and TLR-4 ligands [99,100], indicating that stress and TLR signaling could reciprocally modulate development of NEC. The “omics” data also suggest that inflammatory cytokines and chemokines highlighted in several studies could serve as potential biomarkers for NEC. With the advances in microfluidics and the development of bead-based multiplex arrays, performing a longitudinal high throughput cytokine and chemokine profiling of serum from a cohort of preterm babies could lead to the identification of diagnostic and predictive biomarkers to understand the progression of NEC.
microorganisms-11-02206-t002_Table 2Table 2Gene expression analysis in NEC.ReferenceTechnologyTissueSample SizeDEGsKey GenesKey PathwaysKathy Yuen Yee Chan et al. [71]MicroarrayIntestinal tissue from NEC preterm infantsNEC n = 5Ctrl n = 4↑ 857 ↓ 1285TLR2, TLR4, TREM1, NFkb, AP-1,H1F1AAngiogenesis, Arginine metabolism, cell adhesion, chemotaxis, extracellular matrix remodeling, hypoxia and oxidative stress, inflammation and muscle contraction.Guanglin Chen et al. [72]MicroarraySmall bowel specimens from NEC preterm infantsNEC n = 5Ctrl n = 4↑ 367 ↓ 2262AGT, IL-8, KNG1, ACACB and CATTryptophan, fatty acid, and arachidonic acid metabolismColin MartinMicroarrayParaffin embedded tissue blocks of NEC samplesNEC n = 6Ctrl n = 6↑ 47↓ 37PLA2G2A, H19, AGR2, S100A8, B2M, LOC100132488, CEBPB, LOC643358, LOC100130980, GUCA2A, RARRES1, LOC400963, RPS29, LOC647361, RPS15A, S100A10, LOC100129902, XAF1, TIMP1, SCTR, SERPINA3, LOC389342, EVPL, IFITM2, LOC728937, IFITM3, CEBPD, CLDN15, PPP1R14A, AQP10, REG3G, TUBA1B, REG1B, LOC392437, CREB3L3, C10orf116, ENO1Signal transducer and activator of transcription 3 (STAT3), prolactin (PRL), interlukin-1 beta (IL-1β), signal transducer and activator of transcription 1(STAT1), and interferon gamma (IFNγ)Eric Tremblay et al. [79]RNASeqIntestinal tissue from NEC preterm infantsNEC n = 9Ctrl n = 6↑ 383 ↓ 421CXCL10, TLR4, TLR10, REG3A, DEF5A, DEF6A, LCN2, TFF1, CXCL8, TFF3, BHA2, HBG2Altered T and B cell signaling, B cell development, pattern recognition receptors for bacteria and viruses.Md. Rabiul Auwul et al. [87]RNASeqNEC tissues from preterm infantsNEC n = 9Ctrl n = 5↑ 398↓ 568HBB, HBM, HBZ, ALAS2, HBA1, HBG1, HBA2, ASHP, HBQ1, HBD, IGJ, REG3A, POU2AF1, DEFA5, NEB, TNNT3, TNN11, TNNT1, MYL1Metabolic processes, regulation of immune response, cell communication, and complement cascade.Zhuojun Xie et al. [84]RNASeqNEC tissues, NEC-SC-diagnosed NEC with clinical resection, NOR derived from normal part of ileum in NEC-SC patientsNEC n = 4NEC-SC n = 3Ctrl NOR n = 5NEC vs. NEC-SC↑ 37↓ 7NEC vs. NOR↑ 3465↓ 2499NEC-SC vs. NOR↑ 846↓ 613HBG2, CCN4, IGF2, SOX11, CYP3A4, TEME54, SCIN, PTK6, XIRP1, MMP12, GSTM1, BOLA2B, KDM5D, UTY, AOPB, RPS4Y1, CEMIP, SLC4A1, KRT19, PIGR, and FAM3DToll-like receptor signaling pathways, Th17 cell differentiation and cytokine–cytokine receptor interactions.DEGs = differentially expressed genes, ↑ = upregulated, ↓ = downregulated.

While gene profiling studies have been useful in the identification of many promising molecular targets associated with NEC, many of these studies so far have been limited by a small sample size as well as a lack of validation and functional studies (Table 2). Therefore, the discoveries made so far must be interpreted with caution. Adequate sample size is key to achieving sufficient power for statistical analysis. Given that NEC afflicts only about 5–10% of infants born preterm [101], obtaining sufficient fresh tissues for analysis is challenging. To circumvent this obstacle, as proof of principle, we successfully isolated high-quality RNA from paraffin-embedded tissue blocks to perform transcriptomic analysis. Similarly, Steward et al. utilized existing banks of paraffin-embedded tissues from NEC babies. The authors isolated DNA from tissue blocks and performed 16S rRNA analysis to characterize the microbiome at the site of the disease [102]. We suggest that obtaining formalin-fixed NEC samples from several archival tissue specimens could be one way to increase the sample size for future genetic studies. Further, all gene profiling studies so far are limited to the analysis of NEC and adjacent healthy tissues. As technology advances and allows for a more detailed exploration of the genome, it is expected that whole exome/genome sequencing will uncover novel genes, biomarkers, and pathways in NEC [103]. Until then, increasing sample size and performing in-depth validation studies are required to better interpret and understand the molecular targets that have been identified so far. 

## 7. Conclusions

NEC, a common gastrointestinal disease in premature infants, is associated with high morbidity and mortality. Prematurity, microbial dysbiosis, and enteral feeding are widely accepted risk factors. In recent years, substantial efforts have been made to understand the factors that affect the pathogenesis of NEC. Evidence suggests that maternal prenatal stress and stress-induced microbial dysbiosis can predispose an infant to NEC. Gene expression profile analysis reveals a predominantly altered immune response in the intestine of NEC neonates. Several “omics”-based studies highlight involvement of TLRs and pro-inflammatory cytokines and chemokines in the intestinal tissues of babies with NEC. We conclude that early life exposure to pre- and post-natal stress in babies alters the microbiome, and induces a pro-inflammatory environment that is mediated by dysregulated TLR signaling. Further experiments are needed to investigate the link between stress-induced cortisol and TLR expression and function in the intestinal tissues. With advancements in technology, future genetic studies on NEC hold the potential to improve and lead to the development of biomarkers and novel molecular targets to understand the pathogenesis of NEC.

## Figures and Tables

**Table 1 microorganisms-11-02206-t001:** Psychological stressors.

Stressors
Life changes
Drug, alcohol, and substance abuse
Emotional stresses such as grief, anxiety, depression, or other mental illness
Socioeconomic status
Natural disasters
Nutritional stress (starvation and over-eating)

## Data Availability

Not applicable.

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
