# Peer review of "The Impact of Stress, Microbial Dysbiosis, and Inflammation on Necrotizing Enterocolitis"

_microorganisms, 2023, doi:10.3390/microorganisms11092206_

Round 1

Reviewer 1 Report

This review brings together concepts important to understanding NEC pathophysiology: stress and  microbial dysbiosis. There are new and interesting speculations raised about perinatal stress and paternal stress impact on this disease that are founded in studies. There is a section on microbial dysbiosis in NEC followed by a section describing studies associating stress with microbial dysbiosis that follows logically and is based on contemporary studies. All is well connected and logical until the end section on gene expression profiling in NEC through a variety of omics techniques.. This section does not connect well to the rest of the paper. It provides interesting information that potentially could be linked to stress and microbial dysbiosis and NEC, but the linkages are not explicated.   The conclusion is very brief, not critical,  and does not link the three areas of review.

Author Response

We thank the reviewer for their comments. Their points are well taken. We revised our review accordingly:

1) We linked the role of certain genes highlighted in several omics studies such as TLRs to stress-induced cortisol and microbiome. 

2) We revised the conclusion accordingly. 

Reviewer 2 Report

The manuscript is relevant and well-written.

However, it is recommended to highlight the possible role of Clostridium spp., such as Clostridium sensu stricto 1, and bacterial butyrate in the NEC pathogenesis.

It is also recommended that the names of taxa be corrected in accordance with the reclassification and given in italics. For example, Bacillota, Bacteroidota, Pseudomonadota instead of the obsolete Firmicutes, Bacteroidetes, and Proteobacteria.

Author Response

We thanks the reviewer for the comments. We have addressed all of them:

1) We  highlighted the possible role  Clostridium sensu stricto 1 cited in a study.

2) We also included a section on role of bacterial butyrate in the NEC pathogenesis.

3) We changed the names of taxa as recommended.